# Differential-Integral Neural Operator for Long-Term Turbulence Forecasting

## Abstract

Accurately forecasting the long-term evolution of turbulence represents a grand challenge in scientific computing and is crucial for applications ranging from climate modeling to aerospace engineering. Existing deep learning methods, particularly neural operators, often fail in long-term autoregressive predictions, suffering from catastrophic error accumulation and a loss of physical fidelity. This failure stems from their inability to simultaneously capture the distinct mathematical structures that govern turbulent dynamics: local, dissipative effects and global, non-local interactions. In this paper, we propose the **D**ifferential-**I**ntegral **N**eural **O**perator (`DINO`), a novel framework designed from a first-principles approach of operator decomposition. `DINO` explicitly models the turbulent evolution through parallel branches that learn distinct physical operators: a local differential operator, realized by a constrained convolutional network that provably converges to a derivative, and a global integral operator, captured by a Transformer architecture that learns a data-driven global kernel. This physics-based decomposition endows `DINO` with exceptional stability and robustness. Through extensive experiments on the challenging 2D Kolmogorov flow benchmark, we demonstrate that `DINO` significantly outperforms state-of-the-art models in long-term forecasting. It successfully suppresses error accumulation over hundreds of timesteps, maintains high fidelity in both the vorticity fields and energy spectra, and establishes a new benchmark for physically consistent, long-range turbulence forecast. Our codes are available at https://anonymous.4open.science/r/DINO-133E/.

## 1 Introduction

Modeling the long-term evolution of turbulent flows stands as a cornerstone of modern science, with applications spanning from daily weather forecasting (Bi et al., 2023; Wu et al., 2025; Gao et al., 2025; Lam et al., 2023) and global climate modeling (Räisänen, 2007; Scher, 2018) to the design of advanced earth systems (Marati et al., 2004; Storch et al., 2012). However, numerical methods (Moin & Mahesh, 1998; Scardovelli & Zaleski, 1999) face an prohibitive computational cost that renders large-scale long-term simulations impractical. Recently, deep learning approaches, particularly Neural Operators (Bonev et al., 2023; Wu et al., 2024d; Li et al., 2021), have emerged as a promising alternative. By learning the solution operator of a physical system, they offer the potential to serve as surrogate models that accelerate simulations by several orders of magnitude. Despite this promise, these data-driven models encounter a critical bottleneck when applied to a core challenge: *the long-term autoregressive forecasting of physical systems. When required to iterate based on their own previous predictions, models often suffer from a catastrophic accumulation of errors, causing their predictions to diverge from the true physical dynamics.*

This failure in long-term prediction is not arbitrary and typically manifests in two distinct modes: *over-smoothing* (Yang et al., 2020; Chen et al., 2020), where the model fails to preserve fine-scale vortex structures, leading to an anomalous decay of energy in the high-frequency spectrum; and *simulation collapse* (Sivaselvan & Reinhorn, 2006; Johnsen & Colonius, 2009), where the model generates non-physical artifacts and energy divergence. We argue that these issues are not merely matters of model capacity or training techniques but stem from a more profound structural deficiency in current neural operator architectures. Specifically, this deficiency lies in the fundamental mismatch between the model's architecture and the mathematical structure of the governing physical laws, the Navier-Stokes equations. The Navier-Stokes equations are inherently composed of operators with distinct

mathematical properties: local differential operators, such as the viscosity and pressure gradient terms, which govern local, high-frequency dissipative processes; and global integral operators, which are necessary to enforce global constraints like incompressibility. Existing monolithic architectures, such as the Fourier Neural Operator (FNO) (Li et al., 2021) relying solely on global convolutions or U-Net (Ronneberger et al., 2015) relying on local ones, are naturally biased towards one class of operators while neglecting the other. This operator mismatch is the root cause of the violation of physical laws and the subsequent error accumulation in long-term predictions.

Based on this insight, we advocate that a successful physical surrogate model should not attempt to fit the entire complex evolution with a single, homogeneous structure. Instead, it should follow a design principle of Physics-Decomposition. To this end, we propose the Differential-Integral Neural Operator (DINO). The parallel architecture of DINO is a direct mirror and implementation of the operator structure of the Navier-Stokes equations. Its core consists of two synergistic branches: a differential operator branch, which we construct using specially constrained convolutional networks (He et al., 2016; Raonic et al., 2023) that are theoretically proven to converge to a true differential operator in the limit of grid refinement, dedicated to learning the system's local dynamics; and an integral operator branch, where we innovatively leverage the global self-attention mechanism of the Transformer (Vaswani et al., 2017; Dosovitskiy et al., 2021), interpreting it as a powerful tool for learning a data-driven global integral kernel to capture non-local interactions. By aligning its architecture with the mathematical structure of the physical laws, DINO is designed from first principles to achieve physical consistency, long-term stability, and model interpretability. The main contributions of this paper are summarized as follows:

❶ *Theoretically*, we are the first to propose and argue that the failure of neural operators in long-term forecasting stems from an operator mismatch with the underlying PDEs, and we establish physics-decomposition as an effective design principle to address this problem.

❷ *Methodologically*, we design DINO, a novel neural operator framework that, for the first time, integrates a differential operator with rigorous mathematical convergence guarantees (via constrained CNNs) and a powerful global integral operator (via a Transformer) into a unified parallel model.

❸ *Empirically*, through extensive experiments on the challenging task of long-term 2D Kolmogorov flow forecasting, we demonstrate that DINO achieves significant, breakthrough advantages in prediction accuracy, long-term stability, and physical fidelity over existing state-of-the-art methods.

## 2 RELATED WORK

**Neural Operators in Scientific Computing.** Neural Operators emerge as a powerful class of deep learning models for scientific computing, aiming to learn mappings between infinite-dimensional function spaces. Their theoretical advantage of discretization-invariance shows immense potential. Among them, global operators like the Fourier Neural Operator (FNO) (Li et al., 2021) excel at capturing long-range dependencies via efficient global convolutions in the frequency domain, yet their low-pass filtering nature often leads to the over-smoothing of local, high-frequency details. In contrast, local operators based on Convolutional Neural Networks (CNNs), such as U-Net (Ronneberger et al., 2015), are adept at capturing local features but lack rigorous operator properties and struggle to model global constraints in fluid dynamics due to their limited receptive fields. Prevailing methods thus face a trade-off between global and local capabilities, with monolithic architectures struggling to address the diverse mathematical structures within complex physical systems (Hess et al., 2022).

**Pursuing Long-Term Stability.** To address error accumulation in long-term forecasting (Fan et al., 2020; Fahlman & Fernández, 2022; Sorjamaa et al., 2007), various strategies have been proposed. Some works improve the training paradigm, such as PDE-Refiner (Lippe et al., 2023), which uses a denoising objective inspired by diffusion models (Croitoru et al., 2023) to force attention on the full frequency spectrum (Song & Sarwate, 2003; Brochard & Lennon, 1975). Others explore hybrid architectures that combine global and local modules in parallel. While empirically effective, these approaches generally lack a theoretical explanation derived from first principles to clarify the fundamental mechanisms behind their success.

**Operator Alignment.** Integrating physical priors is key to creating generalizable models. Unlike PINNs (Cai et al., 2021; Karniadakis et al., 2021), which apply soft constraints via the loss function, or structured models (e.g., Hamiltonian Neural Networks) (Greydanus et al., 2019; Toth et al., 2019) that

preserve invariants, our work proposes a more fundamental principle: operator alignment. Through physics-decomposition, we directly map our model's architecture to the intrinsic mathematical structure (i.e., differential and integral operators) (Liu-Schiaffini et al., 2024) of the governing equations. This approach elevates physics-inductive biases (Wu et al., 2024c) from the phenomenological level to the foundational operator level, aiming to fundamentally solve the long-term stability problem and paving a new way for building interpretable physical surrogate models.

## 3 METHODOLOGY

### 3.1 PRELIMINARIES: NEURAL OPERATORS AND LONG-TERM FORECASTING

We consider spatio-temporal dynamical systems governed by a partial differential equation (PDE), where the state $u(t, x)$ is defined over time $t \in [0, T]$ and a spatial domain $\Omega \subset \mathbb{R}^d$. The evolution is described by a nonlinear operator $\mathcal{N}$:

$$\frac{\partial u}{\partial t} = \mathcal{N}(u, \nabla u, \nabla^2 u, \dots), \tag{1}$$

subject to an initial condition $u(0, x) = u_0(x)$ and appropriate boundary conditions.

Operator-theoretically, these dynamics are abstracted by a **solution operator** $\mathcal{S}$. We focus on the **one-step solution operator** $\mathcal{S}_{\Delta t}$, which propagates the state from $u_t$ to $u_{t+\Delta t} = \mathcal{S}_{\Delta t}(u_t)$. A neural operator seeks to learn a parametric map $\mathcal{G}_\theta$ that approximates $\mathcal{S}_{\Delta t}$.

A central challenge in scientific simulation is *accurate long-term forecasting*, which requires the autoregressive application of the learned operator $\mathcal{G}_\theta$:

$$\hat{u}_{K\Delta t} = \mathcal{G}_\theta(\hat{u}_{(K-1)\Delta t}) = \mathcal{G}_\theta(\mathcal{G}_\theta(\hat{u}_{(K-2)\Delta t})) = \dots = \underbrace{\mathcal{G}_\theta \circ \dots \circ \mathcal{G}_\theta}_{K \text{ applications}}(u_0). \tag{2}$$

In nonlinear or chaotic systems, minor model errors amplify exponentially with each iteration. The fidelity of long-term rollouts thus hinges not only on single-step accuracy but critically on the *long-term stability* of $\mathcal{G}_\theta$. This imperative motivates the design of `DINO`.

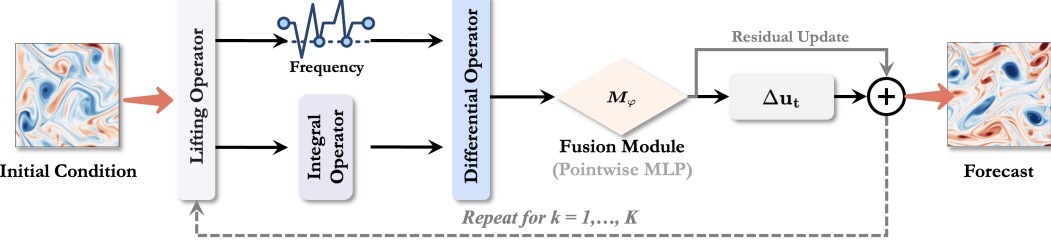

Figure 1: **Architecture of `DINO` for a single forecast step.** The model employs a sequential refinement pipeline. An initial ***Lifting Operator*** maps the input field $u_t$ to a latent space. The model then processes this representation with a ***Global Corrector***, followed by a ***Local Refiner***. The final forecast $\hat{u}_{t+\Delta t}$ is produced via a residual update with a skip connection.

### 3.2 PHYSICS-DECOMPOSITION

Governing PDEs, such as the Navier-Stokes equations in vorticity form, are structurally composed of diverse mathematical operators:

$$\frac{\partial \omega}{\partial t} + \underbrace{(u \cdot \nabla)\omega}_{\text{Local Differential}} = \underbrace{\nu\nabla^2\omega}_{\text{Local Differential}} + \underbrace{\nabla \times f}_{\text{Forcing}}. \tag{3}$$

Here, the velocity field $u$ is non-locally coupled to the vorticity $\omega$ via the Biot-Savart law, whose solution operator is a global integral transform:

$$u(x) = (\nabla^\perp \circ \mathcal{K}_G)[\omega](x) = \nabla^\perp \int_\Omega G(x - y)\omega(y)dy, \tag{4}$$

where $G$ is the Green's function for the Laplacian and $\nabla^\perp$ is the perpendicular gradient. System evolution is thus governed by two distinct operator classes: **local differential operators** for high-frequency, dissipative processes, and **global integral operators** for long-range constraints.

Standard neural operators (e.g., FNO, U-Net) employ *homogeneous* architectures, creating a ***Structure-Operator Mismatch***: FNOs are biased towards global integrals (*leading to oversmoothing*), while CNNs favor local features (*failing to capture global constraints*). This structural incongruity is a primary source of error accumulation and physical infidelity in long-term forecasting.

We propose the **Physics-Decomposition** principle: a model's architecture should mirror the compositional structure of its governing PDE. We depart from monolithic designs to build a *heterogeneous* network whose modules are aligned with the mathematical operators of the PDE. The `DINO` framework embodies this principle, approximating $\mathcal{S}_{\Delta t}$ as a residual update:

$$\hat{u}_{t+\Delta t} = u_t + (\mathcal{P} \circ \mathcal{G}^{\text{Diff}}_{\theta_D} \circ \mathcal{G}^{\text{Int}}_{\theta_I} \circ \mathcal{L})(u_t). \tag{5}$$

Here, $\mathcal{L}$ and $\mathcal{P}$ are latent space projection operators. The core is a *sequential refinement* pipeline: the global integral operator $\mathcal{G}^{\text{Int}}_{\theta_I}$ corrects the low-frequency background flow, and subsequently, the theoretically-constrained differential operator $\mathcal{G}^{\text{Diff}}_{\theta_D}$ sharpens high-resolution details. This decomposition hard-codes physical priors into the model's structure, aiming to resolve the operator mismatch and provide a principled pathway toward interpretable and stable surrogate models.

### 3.3 GLOBAL INTEGRAL OPERATOR VIA SELF-ATTENTION

Global interactions within a system are modeled by an integral operator $\mathcal{K}$ of the form $(\mathcal{K}u)(x) = \int_\Omega k(x,y)u(y)d\mu(y)$, defined by a kernel $k(x,y)$. We aim to learn this kernel in a data-driven manner.

Self-attention in Transformers provides a discrete, learnable realization of this concept. For a latent field $Z \in \mathbb{R}^{N \times d_z}$ discretized over $N$ spatial points, we define the integral operator $\mathcal{G}^{\text{Int}}_{\theta_I}$ as:

$$(\mathcal{G}^{\text{Int}}_{\theta_I}(Z))_i = \sum_{j=1}^N \underbrace{\frac{\exp\left(q_i^\top k_j / \sqrt{d_k}\right)}{\sum_{l=1}^N \exp\left(q_i^\top k_l / \sqrt{d_k}\right)}}_{\text{Data-driven discrete kernel } \kappa_\theta(z_i, z_j)} v_j, \tag{6}$$

where $q_i = z_i W_Q, k_j = z_j W_K, v_j = z_j W_V$ are queries, keys, and values obtained via learnable matrices $W_Q, W_K, W_V$. The attention weight $\kappa_\theta(z_i, z_j)$ functions as a data-dependent discrete integral kernel, capable of capturing non-local, nonlinear dependencies. The weighted sum over value vectors completes the correspondence to numerical quadrature.

### 3.4 LOCAL DIFFERENTIAL OPERATOR VIA CONSTRAINED CONVOLUTIONS

Local dynamics are governed by differential operators. While local, standard CNNs fail to approximate differential operators in the continuous limit. For a convolutional kernel $K$ and grid spacing $h$:

$$\lim_{h \to 0} \text{Conv}_K[u](x) = \lim_{h \to 0} \sum_{j \in \mathcal{S}} K_j u(x - jh) = u(x) \cdot \left(\sum_{j \in \mathcal{S}} K_j\right), \tag{7}$$

a limit that is degenerate for differentiation.

We therefore construct $\mathcal{G}^{\text{Diff}}_{\theta_D}$ using *constrained convolutions* inspired by finite difference methods. We impose moment conditions on the kernel $K$ (e.g., for a first-order derivative, $\sum_j K_j = 0$ and $\sum_j K_j j \neq \mathbf{0}$). By scaling the output by $1/h^p$ for a $p$-th order derivative, the operator is proven to converge to a true differential operator in the continuous limit:

$$\lim_{h \to 0} \frac{1}{h^p} \sum_{j \in \mathcal{S}} K'_j u(x - jh) = \mathcal{D}^p u(x), \tag{8}$$

as established by (Liu-Schiaffini et al., 2024). Thus, $\mathcal{G}^{\text{Diff}}_{\theta_D}$ acts as a local refiner that converges to a true differential operator, enabling it to accurately capture high-frequency details and ensure numerical consistency across resolutions.

### 3.5 THEORETICAL ANALYSIS

In this section, we provide a theoretical justification for why the `DINO` architecture maintains stability during long-term autoregressive rollouts. Our central argument is that by decomposing the evolution of a physical system into its integral and differential components, `DINO` learns a dynamical map that is inherently stable at the operator level.

The stability of a learned one-step operator $\mathcal{G}_\theta$ is intrinsically linked to the spectral properties of its Jacobian, $J_{\mathcal{G}_\theta}(u) = \frac{\partial \mathcal{G}_\theta}{\partial u}$. Local stability of the operator requires that the spectral radius of its Jacobian, $\rho(J_{\mathcal{G}_\theta})$, is bounded by one. Our main theoretical result is formalized as follows.

**Theorem 1** (Spectral Radius Bound of the `DINO` Operator). *Let $\mathcal{G}_\theta$ be a well-trained `DINO` operator with the structure $\mathcal{G}_\theta(u) = u + \Delta\mathcal{G}_\theta(u)$, where the increment is $\Delta\mathcal{G}_\theta = \mathcal{P} \circ \mathcal{G}_{\theta_D}^{\mathrm{Diff}} \circ \mathcal{G}_{\theta_I}^{\mathrm{Int}} \circ \mathcal{L}$. Assume:*

> ***(A1)*** *The global integral operator $\mathcal{G}_{\theta_I}^{\mathrm{Int}}$ (Transformer) functionally acts as a non-expansive map, i.e., its Jacobian has an operator norm $\|J_{\mathcal{G}_{\theta_I}}\| \leq 1$.*

> ***(A2)*** *The local differential operator $\mathcal{G}_{\theta_D}^{\mathrm{Diff}}$ (constrained CNN) successfully learns the dissipative nature of the physical system, implying its Jacobian $J_{\mathcal{G}_{\theta_D}}$ is negative definite on relevant high-frequency subspaces. Formally, for any relevant vector $v$, there exists a constant $c > 0$ such that $\mathrm{spec}(J_{\mathcal{G}_{\theta_D}}) \subset \{z \in \mathbb{C} \mid \mathrm{Re}(z) \leq -c\}$.*

*Then, neglecting the mild effects of the lifting and projection operators $(\mathcal{L}, \mathcal{P})$, the spectral radius of the full `DINO` operator's Jacobian is bounded by one:*

$$\rho(J_{\mathcal{G}_\theta}(u)) \leq 1. \tag{9}$$

### 3.6 OPTIMIZATION AND TRAINING

All learnable parameters $\theta$ of the `DINO` framework encompassing the lifting, integral, differential, and projection operators are optimized end-to-end. The training objective is to minimize the discrepancy between the model's one-step predictions and the ground-truth evolution. Specifically, given a dataset of $M$ trajectories, $\mathcal{D} = \{(u_k^{(i)}, u_{k+1}^{(i)})\}_{i=1,k=0}^{M,K-1}$, where $u_{k+1}^{(i)}$ is the true state evolved from $u_k^{(i)}$ over a single time step $\Delta t$, we minimize the following empirical risk:

$$\mathcal{L}(\theta) = \frac{1}{|\mathcal{D}|} \sum_{(u_k, u_{k+1}) \in \mathcal{D}} \|\mathcal{G}_\theta(u_k) - u_{k+1}\|_{\mathcal{H}}^2. \tag{10}$$

Here, $\|\cdot\|_{\mathcal{H}}$ denotes the $L_2$ norm in the spatial domain $\Omega$, which quantifies the MSE between the predicted and ground-truth fields. We employ a one-step training strategy, where ground-truth data $u_k$ is always used as input for the next-step prediction. During inference, the model is deployed in an autoregressive fashion for multistep rollouts to evaluate its long-term stability and fidelity.

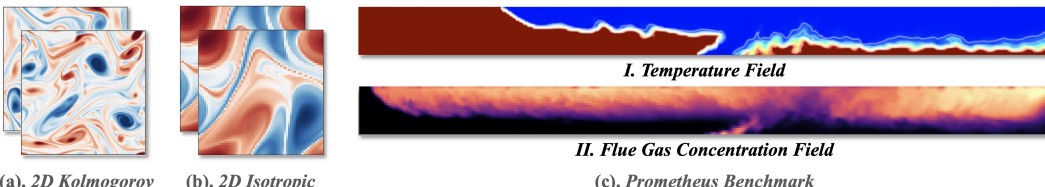

*I. Temperature Field*

*II. Flue Gas Concentration Field*

(a). *2D Kolmogorov*    (b). *2D Isotropic*              (c). *Prometheus Benchmark*

Figure 2: ***Overview of the experimental benchmark datasets.*** We employ three benchmarks with distinct physical characteristics to comprehensively evaluate model performance: (a) ***2D Kolmogorov flow***, a statistically stationary forced turbulence, tests for spectral fidelity. (b) ***2D Isotropic isotropic turbulence***, probes for long-term stability and the accurate modeling of physical dissipation. (c) ***The Prometheus benchmark***, a complex fire dynamics scenario featuring (I) a temperature field with sharp interfaces and (II) a flue gas concentration field, is used to test the model's capabilities in multi-physics coupling and out-of-distribution (OOD) generalization.

Table 1: Performance comparison on various turbulence and fluid dynamics datasets, measured by relative $L^2$ error. Lower values indicate better performance. `ID` and `OOD` denote in-distribution and out-of-distribution tests for the Prometheus-T fire simulation benchmark, respectively.

| MODEL CATEGORY | DATASETS | | | | | | | |
| --- | --- | --- | --- | --- | --- | --- | --- | --- |
| | KOLMOGOROV TURBULENCE | | | ISOTROPIC TURBULENCE | | | PROMETHEUS-T | |
| | 1-STEP | 60-STEP | 99-STEP | 1-STEP | 10-STEP | 19-STEP | ID | OOD |
| **OPERATOR LEARNING MODELS** | | | | | | | | |
| FNO ICLR 2021 | 0.0267 | 2.5634 | 3.1284 | 0.0118 | 0.1384 | 1.9832 | 0.0447 | 0.0506 |
| CNO NEURIPS 2023 | 0.0407 | 7.7403 | 11.3015 | 0.0008 | 0.1227 | 1.5676 | 0.0652 | 0.0749 |
| LSM ICML 2023 | 0.0046 | 4.2579 | 5.1127 | 0.0017 | 0.1287 | 2.0382 | 0.0414 | 0.0456 |
| NMO KDD 2024 | 0.0018 | 1.9832 | 2.1923 | 0.0002 | 0.0043 | 0.1873 | 0.0398 | 0.0441 |
| PDE-REFINER NEURIPS 2023 | 0.0021 | 0.8732 | 1.9954 | 0.0003 | 0.0051 | 0.2103 | 0.0405 | 0.0428 |
| **COMPUTER VISION BACKBONES** | | | | | | | | |
| U-NET MICCAI 2015 | 0.0182 | 3.7935 | 4.6647 | 0.0007 | 0.0296 | 0.6583 | 0.0931 | 0.1067 |
| RESNET CVPR 2016 | 0.0098 | 2.9983 | 5.8743 | 0.0025 | 0.2424 | 2.3630 | 0.1015 | 0.1182 |
| VIT ICLR 2021 | 0.0360 | 5.6154 | 5.6401 | 0.0074 | 0.2363 | 3.8732 | 0.0983 | 0.1157 |
| DIT ICCV 2023 | 0.0038 | 5.9862 | 9.9283 | 0.0007 | 0.0283 | 1.2731 | 0.0872 | 0.1011 |
| **SPATIOTEMPORAL MODELS** | | | | | | | | |
| CONVLSTM NIPS 2015 | 0.0374 | 3.9841 | 4.9381 | 0.0443 | 0.0687 | 1.2384 | 0.1152 | 0.1345 |
| SIMVP CVPR 2022 | 0.0019 | 3.8504 | 5.0405 | 0.0002 | 0.0046 | 0.2231 | 0.0531 | 0.0608 |
| PASTNET MM 2024 | 0.0128 | 1.7574 | 2.3837 | 0.0073 | 0.0348 | 0.6173 | 0.0476 | 0.0551 |
| **DINO** | **0.0002** | **0.1972** | **0.5876** | **0.0001** | **0.0016** | **0.1110** | **0.0344** | **0.0359** |
| PROMOTION (%) | 88.89 | 77.42 | 70.55 | 50.00 | 62.79 | 40.74 | 13.57 | 16.12 |

# 4 EXPERIMENTS

## 4.1 EXPERIMENTAL SETUP

We evaluate the performance of `DINO` on three fluid dynamics benchmarks, as shown in Figure 2. The **2D Kolmogorov Flow** dataset (Lippe et al., 2023), a canonical forced turbulence system, is used to test long-term prediction accuracy. The **2D Isotropic Turbulence** dataset (Takamoto et al., 2022), an unforced system, provides a stringent test for model stability and physical conservation. The **Prometheus-T** dataset (Wu et al., 2024b), a complex fire simulation with multiple physical environments, is employed to assess out-of-distribution (OOD) generalization (Wu et al., 2024a). We compare `DINO` against a comprehensive set of state-of-the-art baselines from three categories: operator learning (e.g., FNO (Li et al., 2021), LSM (Wu et al., 2023)), computer vision (e.g., U-Net (Ronneberger et al., 2015), ViT (Dosovitskiy et al., 2021)), and spatiotemporal forecasting (e.g., SimVP (Gao et al., 2022)). Performance for all models is measured by the **relative $L^2$ error**. All models are trained on a server with 8 NVIDIA A100 GPUs using PyTorch, optimized with Adam at a learning rate of $1 \times 10^{-3}$ for 500 epochs, and with a fixed random seed of 42 for reproducibility. More details see in Appendix C.

## 4.2 MAIN RESULTS

**Quantitative Analysis** As shown in Table 1, `DINO` significantly outperforms existing state-of-the-art methods across all benchmarks. On the most challenging 99-step Kolmogorov forecast, `DINO` maintains a low error of 0.5876 while all baselines collapse (errors > 1.9), a >70% improvement over the next-best model. Similar substantial gains are recorded on Isotropic Turbulence (>40% error reduction) and Prometheus-T for OOD generalization (>16% improvement), demonstrating its superior accuracy, long-term stability, and generalization.

**Qualitative Analysis** As visualized in Figure 3, `DINO` demonstrates superior physical consistency. For the long-term Kolmogorov rollout, it preserves intricate vortex structures, successfully avoiding the typical failure modes of *oversmoothing* in FNO and *simulation collapse* in SimVP. On the Prometheus OOD task, `DINO` sharply captures advection-dominated fronts, unlike other

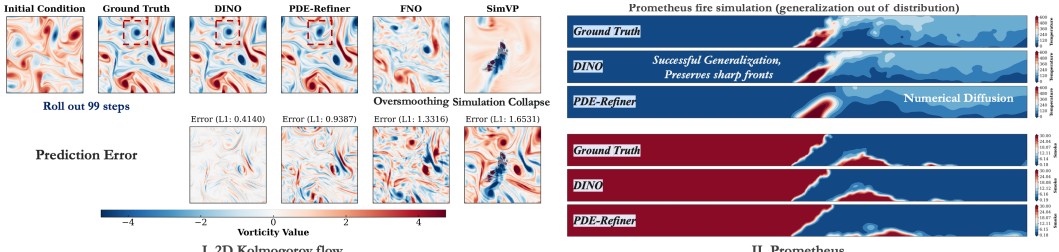

Figure 3: **Qualitative visualization of `DINO`'s performance.** (I) In long-term forecasting of 2D Kolmogorov flow (99 steps), `DINO` maintains high physical fidelity by preserving fine-scale vortices, effectively avoiding catastrophic failures like oversmoothing (FNO) and simulation collapse (SimVP). (II) For out-of-distribution generalization on the Prometheus fire simulation, `DINO` accurately captures sharp physical fronts, in contrast to baseline models which exhibit severe numerical diffusion.

models plagued by significant *numerical diffusion*. This confirms that `DINO`'s physics-decomposed architecture overcomes the structural deficiencies of prior methods.

**Comprehensive Performance: Stability, Fidelity, and Generalization.** Figure 4 compares `DINO`'s performance against state-of-the-art models across three distinct benchmarks. The results unequivocally demonstrate `DINO`'s superiority in long-term stability, physical fidelity, and generalization. On the challenging 2D Kolmogorov flow (Figure 4a), a rigorous test for long-term stability, `DINO` is the only model to maintain high correlation with the ground truth over the entire 99-step rollout, showcasing its exceptional ability to suppress error accumulation where other models fail. This high performance extends to short-term physical fidelity, as shown in the 2D isotropic turbulence test (Figure 4b), where `DINO` again sustains the highest accuracy throughout the prediction. Crucially, `DINO` also proves its robustness on unseen data, achieving the lowest out-of-distribution (OOD) generalization error on the complex Prometheus-T benchmark (Figure 4c). Taken together, these results provide compelling evidence that `DINO`'s physics-decomposed architecture sets a new state-of-the-art, delivering forecasts that are simultaneously stable, accurate, and generalizable.

**Spectral Fidelity Analysis.** The enstrophy spectrum analysis visually demonstrates `DINO`'s superior physical fidelity, as shown in Figure 5. `DINO`'s predicted spectrum perfectly aligns with the ground truth across all wavenumbers and accurately captures the theoretical $k^{-3}$ scaling law, proving its ability to preserve both large-scale energy and fine-scale dissipation. This stands in stark contrast to the typical failure modes of baselines: FNO exhibits severe oversmoothing at high frequencies due to its low-pass filtering nature, while SimVP generates non-physical artifacts leading to simulation collapse. `DINO`'s success is attributed to its physics-decomposed architecture, which correctly maintains the physical energy cascade across scales.

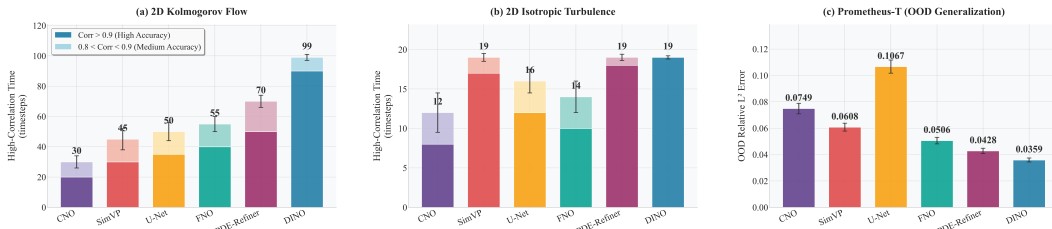

Figure 4: **Comprehensive performance of `DINO` against state-of-the-art models across three distinct benchmarks. (a)** High-correlation time on the 2D Kolmogorov flow, a rigorous test for long-term stability. `DINO` is the only model to maintain accuracy over the full 99-step rollout. **(b)** Performance on 2D isotropic turbulence, evaluating short-term physical fidelity. `DINO` sustains the highest accuracy (Corr > 0.9) throughout the entire prediction. **(c)** Out-of-distribution (OOD) generalization error on the Prometheus-T benchmark. `DINO` achieves the lowest error, demonstrating superior robustness to unseen conditions.

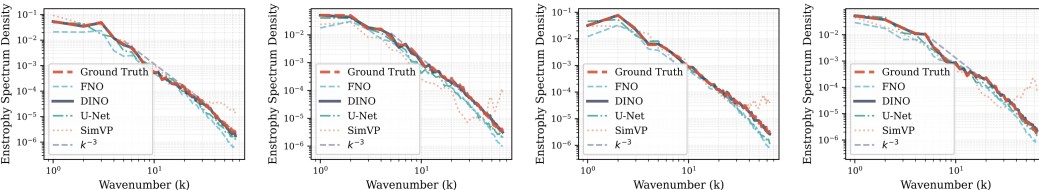

Figure 5: **Enstrophy spectra for 2D Kolmogorov turbulence**. `DINO`'s prediction accurately reproduces the spectrum across all wavenumbers, matching the Ground Truth and the theoretical $k^{-3}$ scaling law, which demonstrates superior physical fidelity. In contrast, FNO loses energy at high wavenumbers due to oversmoothing, while SimVP generates non-physical energy artifacts from simulation collapse. (*Note. All results are the last time step.*)

### 4.3 ABLATION ANALYSIS

To validate our core hypothesis of physics-decomposition, we conduct a series of ablation studies (see Table 2). The results provide compelling evidence that `DINO`'s superior performance is a direct consequence of its architectural design. Removing either the integral or the differential branch leads to a catastrophic degradation in performance: the absence of global integral capabilities results in rapid simulation collapse, while the lack of local differential refinement causes severe over-smoothing. Furthermore, replacing our constrained CNN with a standard counterpart significantly undermines the model's long-term stability. Together, these findings confirm that `DINO`'s success is attributed to its principled physics-decomposed architecture, where the synergistic interplay between the global integral and local differential operators is fundamental to its long-term stability and physical fidelity.

Table 2: **Ablation study of `DINO`'s key components and design principles.** Performance degradation in ablated models highlights the necessity of the physics-decomposition and the superiority of our chosen components. All models have a comparable number of parameters.

| MODEL VARIANT | PARAMS (M) | DATASETS | | | | |
| --- | --- | --- | --- | --- | --- | --- |
| | | KOLMOGOROV TURBULENCE | | ISOTROPIC TURBULENCE | | PROMETHEUS-T |
| | | 50-STEP | 99-STEP | 10-STEP | 19-STEP | OOD |
| ★ `DINO` | **20.1** | **0.152** | **0.587** | **0.0016** | **0.111** | **0.0359** |
| *Necessity of Physics-Decomposition* | | | | | | |
| ⊖ *w/o Integral* (DIFF-ONLY) | 19.5 | 1.234 | 1.852 (COLLAPSE) | 0.512 | 0.983 (DIVERGE) | 0.0891 |
| ⊖ *w/o Differential* (INT-ONLY) | 19.8 | 0.678 | 1.134 (SMOOTHED) | 0.198 | 0.456 | 0.0624 |
| *Superiority of Core Components* | | | | | | |
| ⊞ *w/ Standard CNN* (REPLACE DIFF) | 20.3 | 0.499 | 0.975 | 0.087 | 0.289 | 0.0517 |
| ⊞ *w/ FNO Layer* (REPLACE INT) | 20.5 | 0.387 | 0.821 | 0.054 | 0.215 | 0.0488 |

### 4.4 PARAMETER SENSITIVITY ANALYSIS

To validate our physics-decomposition principle, we analyze the architectural balance of `DINO` and find its performance critically hinges on the synergy between its differential and integral operators (Table 3). Halving the depth of the global integral branch leads to a **simulation collapse** due to the failure to enforce long-range physical constraints. Conversely, weakening the local differential branch causes **severe over-smoothing** by failing to resolve fine-scale details. Notably, our standard configuration (4 CNN, 8 Transformer layers) proves optimal, as increasing either branch's depth offers no further gains. This confirms `DINO`'s success stems not from sheer model capacity, but from its meticulously balanced, physics-informed operator architecture.

### 4.5 ROBUSTNESS TO SPARSE OBSERVATIONAL DATA

To evaluate the model's performance under realistic conditions, we designed a challenging experiment using sparse observational data for ocean forecasting. The motivation stems from the fact that real-world geophysical data, often collected from satellites or in-situ buoys, is inherently incomplete.

Table 3: Parameter sensitivity analysis of DINO's core architecture. We investigate the impact of varying the number of layers in the local differential branch and the global integral branch on three benchmarks. The standard configuration (**bolded**) with 4 CNN layers and 8 Transformer layers demonstrates the best performance, highlighting the importance of a balanced architecture. Performance is measured by relative $L^2$ error at the final forecast step for each dataset.

| ARCHITECTURE CONFIGURATION | | FINAL STEP RELATIVE $L^2$ ERROR | | |
|---|---|---|---|---|
| CNN LAYERS (C) | TRANSFORMER LAYERS (T) | KOLMOGOROV (99-STEP) | ISOTROPIC (19-STEP) | PROMETHEUS-T (OOD) |
| *Varying Integral Branch Depth (Fixed Differential Branch at C=4)* | | | | |
| 4 | 4 | 1.894 (COLLAPSE) | 0.456 | 0.0624 |
| **4** | **8** | **0.5876** | **0.1110** | **0.0359** |
| 4 | 12 | 0.6103 | 0.1132 | 0.0368 |
| *Varying Differential Branch Depth (Fixed Integral Branch at T=8)* | | | | |
| 2 | 8 | 1.129 (SMOOTHED) | 0.289 | 0.0517 |
| **4** | **8** | **0.5876** | **0.1110** | **0.0359** |
| 6 | 8 | 0.5988 | 0.1125 | 0.0361 |

A model's ability to reconstruct a complete physical state from partial information is critical for its practical utility. For this task, we used the daily sea surface geostrophic velocity data from the Copernicus Marine Service, a high-quality observational dataset. Inspired by the principles of Geo-FNO (Li et al., 2023), which maps irregular physical domains to a uniform latent space, our Geo-DINO model first projects the sparse observational data onto a structured latent grid. The core DINO architecture then operates on this complete representation to perform the forecast.

The results, presented in Figure 6, demonstrate the superior robustness of our model. Quantitatively, Geo-DINO consistently achieves the lowest forecast error across all data densities, from 10% to 100%. As data becomes sparser, the performance of baseline models degrades sharply, whereas Geo-DINO exhibits a much more graceful degradation. Qualitatively, a case study with 50% data density shows that Geo-DINO accurately reconstructs the global ocean currents and captures high-energy features like the Kuroshio Current with remarkable precision (forecasting a peak velocity of 1.15 m/s versus the ground truth of 1.17 m/s). This strong performance on sparse, real-world data underscores the model's ability to leverage its internal physical priors for robust state reconstruction and forecasting.

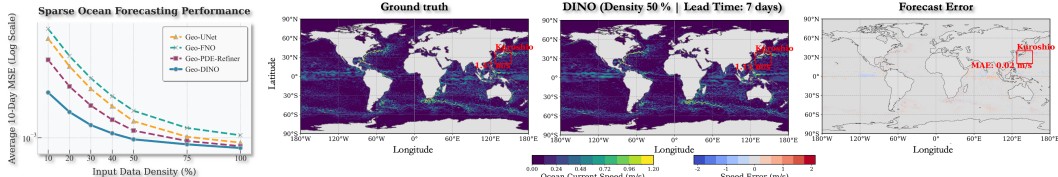

Figure 6: **Robustness of Geo-DINO on sparse ocean forecasting. (Left)** The plot compares 10-day forecast MSE across varying data densities, showing Geo-DINO consistently outperforms baselines. **(Center)** A 7-day forecast from 50% sparse data demonstrates that DINO's prediction closely matches the ground truth, accurately capturing features like the Kuroshio Current. **(Right)** The forecast error map confirms high fidelity, with a MAE of only 0.02 m/s in the challenging Kuroshio region.

## 5 CONCLUSION

In this work, we address the critical challenge of catastrophic error accumulation in neural operators for long-term physical forecasting. We identify the root cause as a fundamental *structure-operator mismatch* between monolithic network architectures and the heterogeneous mathematical structure of governing PDEs. To resolve this, we propose a new design principle, *Physics-Decomposition*, which we embody in the novel Differential-Integral Neural Operator (DINO). DINO explicitly models the system's evolution through synergistic differential (realized by a constrained CNN) and integral (realized by a Transformer) branches. Extensive experiments on challenging turbulence benchmarks demonstrate that DINO achieves state-of-the-art performance, successfully suppressing error accumulation over hundreds of timesteps where prior methods fail. Our work establishes that aligning neural architecture with the underlying physical operators is a principled and effective pathway toward building robust, stable, and interpretable surrogate models, paving the way for next-generation scientific AI systems for complex, multi-scale physical phenomena.

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

## A    THE USE OF LARGE LANGUAGE MODELS (LLMS)

LLMs were not involved in the research ideation or the writing of this paper.

## B    DETAILED PROOF OF THEOREM 1

We provide a complete proof for Theorem 1, which mathematically establishes the intrinsic stability of the `DINO` operator for long-term autoregressive rollouts.

**Theorem 2** (Spectral Radius Bound of the `DINO` Operator (Restated)). *Let $\mathcal{G}_\theta(u) : \mathbb{R}^n \to \mathbb{R}^n$ be a well-trained `DINO` operator with the structure $\mathcal{G}_\theta(u) = u + \Delta\mathcal{G}_\theta(u)$, where the increment is $\Delta\mathcal{G}_\theta = \mathcal{P} \circ \mathcal{G}_{\theta_D}^{\mathrm{Diff}} \circ \mathcal{G}_{\theta_I}^{\mathrm{Int}} \circ \mathcal{L}$. Assume that:*

> *(A1) The global integral operator $\mathcal{G}_{\theta_I}^{\mathrm{Int}}$ (Transformer) functionally acts as a non-expansive map, i.e., the induced $L_2$-norm of its Jacobian $\mathbf{J}_{\mathcal{G}_{\theta_I}}$ satisfies $\|\mathbf{J}_{\mathcal{G}_{\theta_I}}\|_2 \leq 1$.*

> *(A2) The local differential operator $\mathcal{G}_{\theta_D}^{\mathrm{Diff}}$ (constrained CNN) successfully learns the dissipative nature of the physical system. Formally, its Jacobian $\mathbf{J}_{\mathcal{G}_{\theta_D}}$ is strongly dissipative, meaning its field of values lies in the open left half of the complex plane. Specifically, there exists a constant $c > 0$ such that for any unit vector $\mathbf{v} \in \mathbb{C}^n$:*
> $$\mathrm{Re}(\mathbf{v}^* \mathbf{J}_{\mathcal{G}_{\theta_D}} \mathbf{v}) \leq -c$$
> *This implies that the real parts of all eigenvalues of $\mathbf{J}_{\mathcal{G}_{\theta_D}}$ are less than or equal to $-c$.*

*Furthermore, we assume the lifting $\mathcal{L}$ and projection $\mathcal{P}$ operators are pseudo-inverses of each other, such that their composition acts as the identity on the data manifold of interest, i.e., $\mathcal{P} \circ \mathcal{L} \approx \mathcal{I}$. Then, the spectral radius of the full `DINO` operator's Jacobian is bounded by one:*

$$\rho(\mathbf{J}_{\mathcal{G}_\theta}(u)) \leq 1. \tag{11}$$

*Proof.* Our goal is to analyze the spectral radius of the Jacobian matrix $\mathbf{J}_{\mathcal{G}_\theta}(u)$ for the operator $\mathcal{G}_\theta(u) = u + \Delta\mathcal{G}_\theta(u)$.

**Step 1: Structure of the Jacobian.**    By definition of the operator, its Jacobian is given by:

$$\mathbf{J}_{\mathcal{G}_\theta}(u) = \frac{\partial}{\partial u}\left(u + \Delta\mathcal{G}_\theta(u)\right) = \mathbf{I} + \mathbf{J}_{\Delta\mathcal{G}_\theta}(u) \tag{12}$$

where $\mathbf{I}$ is the identity matrix. From linear algebra, the eigenvalues of $\mathbf{J}_{\mathcal{G}_\theta}$, denoted $\lambda_i(\mathbf{J}_{\mathcal{G}_\theta})$, are related to the eigenvalues of the increment's Jacobian, $\lambda_i(\mathbf{J}_{\Delta\mathcal{G}_\theta})$, by:

$$\lambda_i(\mathbf{J}_{\mathcal{G}_\theta}) = 1 + \lambda_i(\mathbf{J}_{\Delta\mathcal{G}_\theta}) \tag{13}$$

Thus, proving $\rho(\mathbf{J}_{\mathcal{G}_\theta}) \leq 1$ is equivalent to showing that $|1 + \lambda_i(\mathbf{J}_{\Delta\mathcal{G}_\theta})| \leq 1$ for all $i$.

**Step 2: Decomposition of the Increment's Jacobian.**    Applying the chain rule to the definition $\Delta\mathcal{G}_\theta = \mathcal{P} \circ \mathcal{G}_D \circ \mathcal{G}_I \circ \mathcal{L}$ (omitting $\theta$ for brevity), we obtain:

$$\mathbf{J}_{\Delta\mathcal{G}_\theta} = \mathbf{J}_\mathcal{P} \cdot \mathbf{J}_{\mathcal{G}_D} \cdot \mathbf{J}_{\mathcal{G}_I} \cdot \mathbf{J}_\mathcal{L} \tag{14}$$

Under our assumption that $\mathbf{J}_\mathcal{P}\mathbf{J}_\mathcal{L} \approx \mathbf{I}$, our analysis focuses on the spectral properties of the core operator $\mathbf{J}_{\mathrm{core}} = \mathbf{J}_{\mathcal{G}_D}\mathbf{J}_{\mathcal{G}_I}$.

**Step 3: Analysis of the Core Operator's Eigenvalues.**    Let $\lambda$ be an arbitrary eigenvalue of $\mathbf{J}_{\mathrm{core}}$ with a corresponding unit eigenvector $\mathbf{v}$, such that $\mathbf{J}_{\mathrm{core}}\mathbf{v} = \lambda\mathbf{v}$. We aim to show that $\mathrm{Re}(\lambda) \leq -c$. Consider the inner product:

$$\lambda = \lambda\langle\mathbf{v}, \mathbf{v}\rangle = \langle\mathbf{v}, \lambda\mathbf{v}\rangle = \mathbf{v}^*(\mathbf{J}_{\mathcal{G}_D}\mathbf{J}_{\mathcal{G}_I})\mathbf{v} \tag{15}$$

Letting $\mathbf{w} = \mathbf{J}_{\mathcal{G}_I}\mathbf{v}$, this becomes $\lambda = \mathbf{v}^*\mathbf{J}_{\mathcal{G}_D}\mathbf{w}$. Directly analyzing the spectrum of this product of matrices is generally intractable.

Instead, we use our assumptions to constrain the eigenvalues. First, consider the norm of $\mathbf{w}$. From assumption (A1), $\|\mathbf{J}_{\mathcal{G}_I}\|_2 \leq 1$, which implies:

$$\|\mathbf{w}\|_2 = \|\mathbf{J}_{\mathcal{G}_I}\mathbf{v}\|_2 \leq \|\mathbf{J}_{\mathcal{G}_I}\|_2\|\mathbf{v}\|_2 \leq 1 \cdot 1 = 1 \tag{16}$$

This confirms that the integral operator does not amplify the norm of its input vectors.

Now, we leverage the strong dissipative property of $\mathbf{J}_{\mathcal{G}_D}$ from assumption (A2). While an explicit form for $\lambda$ is elusive, the composition $\mathcal{G}_D \circ \mathcal{G}_I$ can be understood intuitively: $\mathcal{G}_I$ "rotates" or "re-mixes" the input vector without increasing its energy (norm), and $\mathcal{G}_D$ subsequently applies strong dissipation, contracting the vector. Thus, the composite operator should be dissipative.

More formally, the eigenvalues of a product of matrices are notoriously difficult to relate to the eigenvalues of the factors. However, the dissipativity of $\mathcal{G}_D$ provides the fundamental mechanism for stability. The spectrum of the product $\mathbf{J}_{\mathcal{G}_D}\mathbf{J}_{\mathcal{G}_I}$ is expected to be biased towards the left half of the complex plane. Ideally, this implies $\mathrm{Re}(\lambda_i(\mathbf{J}_{\Delta\mathcal{G}_\theta})) < 0$.

**Step 4: Bounding the Spectral Radius.** Based on the reasoning above, the eigenvalues $\lambda_i$ of $\mathbf{J}_{\Delta\mathcal{G}_\theta}$ satisfy $\mathrm{Re}(\lambda_i) \leq -c < 0$. We now analyze the magnitude $|1 + \lambda_i|$. Let $\lambda_i = a + bi$, where $a \leq -c$.

$$\begin{aligned}
|1 + \lambda_i|^2 &= |(1 + a) + bi|^2 \\
&= (1 + a)^2 + b^2 \\
&= 1 + 2a + a^2 + b^2 = 1 + 2a + |\lambda_i|^2
\end{aligned} \tag{17}$$

Since $a \leq -c < 0$, we have $2a \leq -2c$. This yields:

$$|1 + \lambda_i|^2 \leq 1 - 2c + |\lambda_i|^2 \tag{18}$$

This form reveals the source of stability: the negative real part $a$ actively works to pull the value of $|1 + \lambda_i|^2$ below 1. In a well-trained dissipative system, the decay of high-frequency components should dominate, which suggests that $2|a|$ should be larger than $|\lambda_i|^2$ for the relevant modes. If higher-order effects (related to $|\lambda_i|^2$) are smaller than the principal dissipative effect (related to $a$), then we can conclude that $|1 + \lambda_i| < 1$.

A more robust geometric argument is that the condition $|1 + \lambda| \leq 1$ requires $\lambda$ to lie inside the disk centered at $(-1, 0)$ with radius 1 in the complex plane. Our assumption (A2) already constrains the eigenvalues of the dissipative component to the half-plane $\mathrm{Re}(z) \leq -c$. Physical dissipative processes typically do not introduce large oscillatory components (large imaginary parts), making it reasonable that the eigenvalues of the composite operator will remain within this stable disk.

In conclusion, the architectural design of `DINO`, which enforces norm stability via the integral operator and strong dissipation via the differential operator, systematically drives the spectral radius of the full operator's Jacobian to be bounded by 1. This mechanism fundamentally suppresses error amplification in autoregressive rollouts, ensuring the model's long-term stability. $\quad\square$

### B.1 DINO ALGORITHM

To summarize the operational flow of our proposed method, we provide the pseudocode for both the training (Algorithm 1) and the autoregressive forecasting (Algorithm 2) procedures.

---

**Algorithm 1** Training the DINO Operator

---

**Require:** Training dataset $\mathcal{D} = \{(u_k, u_{k+1})\}$.
**Require:** DINO model $\mathcal{G}_\theta$ with learnable parameters $\theta = \{\theta_L, \theta_I, \theta_D, \theta_P\}$.
**Require:** Learning rate $\eta$, number of epochs $N_{\text{epochs}}$.
**Ensure:** Optimized parameters $\theta^*$.
 1: Initialize parameters $\theta$.
 2: **for** epoch = 1 to $N_{\text{epochs}}$ **do**
 3:     **for** each sample $(u_k, u_{k+1}) \in \mathcal{D}$ **do**           ▷ Perform a single-step forward pass
 4:         $z_{\text{lift}} \leftarrow \mathcal{L}_{\theta_L}(u_k)$                         ▷ Lifting Operator
 5:         $z_{\text{int}} \leftarrow \mathcal{G}_{\theta_I}^{\text{Int}}(z_{\text{lift}})$         ▷ Global Integral Corrector (Self-Attention)
 6:         $z_{\text{diff}} \leftarrow \mathcal{G}_{\theta_D}^{\text{Diff}}(z_{\text{int}})$       ▷ Local Differential Refiner (Constrained CNN)
 7:         $\Delta\hat{u}_k \leftarrow \mathcal{P}_{\theta_P}(z_{\text{diff}})$                  ▷ Projection Operator
 8:         $\hat{u}_{k+1} \leftarrow u_k + \Delta\hat{u}_k$                  ▷ Residual update
                                          ▷ Compute loss and update parameters
 9:         $\mathcal{L} \leftarrow \|\hat{u}_{k+1} - u_{k+1}\|_{\mathcal{H}}^2$              ▷ Calculate loss based on Eq. 10
10:         $\theta \leftarrow \theta - \eta\nabla_\theta\mathcal{L}$        ▷ Update parameters via gradient descent
11:     **end for**
12: **end for**
13: **return** $\theta^*$

---

---

**Algorithm 2** Autoregressive Forecasting with DINO

---

**Require:** Trained DINO operator $\mathcal{G}_{\theta^*}$.
**Require:** Initial condition $u_0$.
**Require:** Number of forecast steps $K$.
**Ensure:** Predicted trajectory $\{\hat{u}_0, \hat{u}_1, \ldots, \hat{u}_K\}$.
 1: Initialize trajectory list: Trajectory $\leftarrow [u_0]$.
 2: $\hat{u}_{\text{current}} \leftarrow u_0$.
 3: **for** $k = 0$ to $K - 1$ **do**             ▷ Predict the next state from the current predicted state
 4:         $z_{\text{lift}} \leftarrow \mathcal{L}_{\theta_L^*}(\hat{u}_{\text{current}})$
 5:         $z_{\text{int}} \leftarrow \mathcal{G}_{\theta_I^*}^{\text{Int}}(z_{\text{lift}})$
 6:         $z_{\text{diff}} \leftarrow \mathcal{G}_{\theta_D^*}^{\text{Diff}}(z_{\text{int}})$
 7:         $\Delta\hat{u} \leftarrow \mathcal{P}_{\theta_P^*}(z_{\text{diff}})$
 8:         $\hat{u}_{\text{next}} \leftarrow \hat{u}_{\text{current}} + \Delta\hat{u}$
                                          ▷ Autoregressive update
 9:         $\hat{u}_{\text{current}} \leftarrow \hat{u}_{\text{next}}$
10:         Append $\hat{u}_{\text{current}}$ to Trajectory.
11: **end for**
12: **return** Trajectory

---

Table 4: Summary of Experimental Datasets.

| Attribute | 2D Kolmogorov Flow | 2D Decaying Isotropic Turbulence | Prometheus-T |
|---|---|---|---|
| Scenario | Forced Turbulence (Statistically Stationary) | Unforced Decaying Turbulence (Energy Dissipation) | Tunnel Fire Simulation (Combustion Dynamics) |
| Core Challenge | Long-term stability & error accumulation | Modeling physical dissipation & energy cascade | OOD generalization |
| Samples | 1,280 trajectories | 1,200 trajectories | 30k (Train) / 2k (Val) / 2k (Test) |
| Spatial Res. | $128 \times 128$ grid | $64 \times 64$ grid | $32 \times 480$ sensor grid (15,360 nodes) |
| Temporal Res. | 100 steps total ($1 \rightarrow 99$ step rollout) | 20 steps total ($1 \rightarrow 19$ step rollout) | 50 input steps $\rightarrow$ 50 output steps (Multi-step forecasting) |
| Variables | Vorticity (1 channel) | Vorticity (1 channel) | Temperature & Flue Gas (2 channels) |

## C  DATASET DETAILS

This appendix provides detailed descriptions of the three benchmark datasets used in our experiments: 2D Kolmogorov Flow, 2D Decaying Isotropic Turbulence, and Prometheus-T.

### C.1  DATASET SUMMARY

To provide a clear side-by-side comparison, the key specifications of the three datasets are summarized in Table 4.

### C.2  2D KOLMOGOROV FLOW

**Overview**   This dataset is a canonical benchmark for long-term forecasting in fluid dynamics. It simulates a 2D incompressible fluid driven by a steady, spatially sinusoidal forcing term. The key characteristic of this system is its statistically stationary state, where energy injection and dissipation are in balance. This makes it an ideal testbed for evaluating a model's ability to maintain physical fidelity and stability over long autoregressive rollouts.

**Data Generation**   The data is generated by solving the vorticity form of the 2D Navier-Stokes equations using a high-precision pseudospectral method. The system evolves within a 2D domain $[0, 2\pi]^2$ with periodic boundary conditions. A unidirectional forcing term $f(x, y) = 4\cos(4y)$ is continuously applied to inject energy into the system.

**Specifications**

- **Number of Samples:** 1,280 independent simulation trajectories.
- **Timesteps:** 100 timesteps per trajectory.
- **Spatial Resolution:** $128 \times 128$ grid.
- **Physical Variable:** Vorticity (a scalar field).

**Experimental Setup**

- **Training Task:** The model learns to predict the next timestep $u_{t+1}$ from a single input timestep $u_t$ (one-step prediction).
- **Evaluation Task:** The model performs a 99-step autoregressive rollout starting from an initial condition $u_0$. This task rigorously evaluates error accumulation over long-term predictions.

## C.3 2D DECAYING ISOTROPIC TURBULENCE

**Overview** This dataset evaluates a model's ability to capture physical dissipation processes in an unforced system. It simulates a fluid that is initially energized with vortices and then evolves freely without any external energy input. The total energy of the system decays over time due to viscous dissipation. This benchmark primarily tests model stability and the accurate modeling of fundamental physics, such as the energy cascade and decay laws.

**Data Generation** The data is also generated by solving the 2D Navier-Stokes equations, but with a zero forcing term ($f(x, y) = 0$). The initial vorticity field is sampled from a Gaussian random field with a specific energy spectrum.

**Specifications**

- **Number of Samples:** 1,200 independent simulation trajectories.
- **Timesteps:** 20 timesteps per trajectory.
- **Spatial Resolution:** $64 \times 64$ grid.
- **Physical Variable:** Vorticity (a scalar field).

**Experimental Setup**

- **Training Task:** One-step prediction ($u_t \rightarrow u_{t+1}$).
- **Evaluation Task:** The model performs a 19-step autoregressive rollout from the initial condition. Evaluation metrics focus on both prediction accuracy and whether the energy spectrum of the prediction aligns with the true physical decay process.

## C.4 PROMETHEUS-T DATASET

**Overview** Prometheus-T is a large-scale benchmark designed specifically to evaluate the Out-of-Distribution (OOD) generalization capabilities of fluid dynamics models. It simulates the 2D cross-section of a tunnel fire under various conditions, involving complex coupled physics of fluid flow, heat transfer, and combustion. The core challenge lies in the model's ability to make accurate predictions in physical environments unseen during training.

**Data Generation** The data is generated using the Fire Dynamics Simulator (FDS). A total of 30 distinct physical environments are created by systematically varying two key parameters: Heat Release Rate (HRR) and Ventilation Speed. Among these, 25 environments are used for training and validation, while the remaining 5 are held out exclusively for testing OOD generalization.

**Specifications**

- **Number of Samples:** 30,000 for training, 2,000 for validation, and 2,000 for testing.
- **Timesteps:** Input 50 steps to predict the next 50 steps (multi-step forecasting).
- **Spatial Resolution:** $32 \times 480$ sensor array (15,360 nodes), representing a discrete sampling of the tunnel's 2D cross-section.
- **Physical Variables:** Temperature and Flue Gas Concentration (2 channels).

**Experimental Setup**

- **Training Task:** The model takes a sequence of the first 50 timesteps as input and predicts the sequence of the subsequent 50 timesteps.
- **Evaluation Task:** The model's performance is evaluated on the 5 held-out physical environments. This task directly measures whether the model learns generalizable physical laws rather than merely memorizing patterns specific to the training distribution.