# OpenReview forum: "Differential-Integral Neural Operator for Long-Term Turbulence Forecasting"
_ICLR.cc/2026/Conference — Submitted to ICLR 2026_

### Official Review · Reviewer_Ph2a · 2025-10-26

**Soundness:** 2
**Presentation:** 3
**Contribution:** 2
**Rating:** 4
**Confidence:** 4

**Summary:**

This paper introduces the Differential-Integral Neural Operator (DINO) for long-term turbulence forecasting. The authors argue that existing neural operators fail at long-term predictions because of a fundamental "structure-operator mismatch" monolithic architectures can't simultaneously capture both the local differential operators (like viscosity terms) and global integral operators (like incompressibility constraints) that govern fluid dynamics. DINO addresses this by explicitly decomposing the architecture into two parallel branches: a differential operator branch using constrained CNNs that provably converge to true derivatives, and an integral operator branch using Transformers to learn global kernels. The authors validate DINO on three benchmarks—2D Kolmogorov flow, 2D isotropic turbulence, and the Prometheus fire simulation dataset showing substantial improvements in long-term stability (>70% error reduction at 99 timesteps) and physical fidelity compared to baselines like FNO, U-Net, and various spatiotemporal models.

**Strengths:**

I think the authors have identified a genuinely interesting problem with current neural operators the idea that architectural homogeneity causes a mismatch with the heterogeneous mathematical structure of PDEs is compelling and well-articulated. The "physics-decomposition" principle makes intuitive sense: if the Navier-Stokes equations naturally separate into local differential terms and global integral terms, why should we force a single architecture to learn both? The use of constrained CNNs with theoretical convergence guarantees to differential operators is a nice touch that adds rigor, and interpreting Transformer self-attention as a learnable integral kernel is creative. The experimental results are honestly quite impressive on the 99-step Kolmogorov rollout, DINO achieves 0.5876 error while all baselines completely collapse with errors above 1.9. That's a massive improvement. The visualizations in Figures 3-5 clearly show DINO avoiding the typical failure modes of oversmoothing and simulation collapse, and the spectral analysis demonstrates it actually preserves the correct k^-3 energy cascade. Good job including the ablation studies and parameter sensitivity analysis these convincingly show that both branches are necessary.

**Weaknesses:**

Okay, so while the results look great, I'm honestly not fully convinced by some of the theoretical justifications and design choices. First, the theoretical analysis in Section 3.5 and Appendix B feels hand-wavy. Theorem 1 claims the spectral radius is bounded by 1, but the proof makes several questionable assumptions. Assumption A1 says the Transformer acts as a non-expansive map with Jacobian norm ≤1, but is this actually true for trained Transformers with self-attention? Self-attention can definitely amplify certain patterns, and there's no architectural constraint enforcing this. Similarly, A2 assumes the constrained CNN learns a strongly dissipative operator with all eigenvalues having real part ≤ -c, but again, there's no guarantee this happens during training. The proof essentially says "if these nice properties hold, then stability follows," but doesn't prove that DINO actually learns operators with these properties. The step from equation (17) to (18) where they argue |1+λ|² ≤ 1 requires 2|a| > |λ|², which they justify by saying "higher-order effects should be smaller," but this is just speculation without any empirical verification or tighter bounds.

The sequential ordering of operators (integral then differential) seems arbitrary and isn't well justified. The authors claim the integral operator "corrects the low-frequency background flow" and then differential "sharpens high-resolution details," but why not the reverse order? Or why not iterate between them multiple times? They show in Table 3 that this specific ordering works, but there's no analysis of alternative architectures. Also, the constrained CNN design is borrowed from prior work (Liu-Schiaffini et al., 2024) and the convergence guarantee only holds in the continuous limit as grid spacing goes to zero but they're working with fixed finite grids (128×128, 64×64), so how good is this approximation in practice? Also a huge red flag is the Liu-Schiaffini et al. paper that explicitly proposes architectures that integrate both local differential operators (via constrained convolutions) and global integral operators (via various mechanisms) for solving PDEs. This is basically the exact same "physics-decomposition" principle that DINO claims to introduce. Yet there's zero experimental comparison to this method in the paper. Not in Table 1, not anywhere. This is a huge red flag.

**Questions:**

Some major questions I had!

1. **Why is there no experimental comparison to Liu-Schiaffini et al. (2024)?** You cite this paper multiple times and borrow the constrained CNN technique from it, but their work also proposes combining localized differential kernels with integral kernels for neural operators which sounds exactly like your "physics-decomposition" principle. Can you run their method on your three benchmarks and include it in your comparison tables? Without this, I can't tell if your improvements come from the general idea of combining local and global operators (which they already proposed) or from your specific implementation choices.

2. **What happens if you reverse the operator ordering or use parallel branches instead of sequential?** You apply the integral operator first, then the differential operator, claiming this corrects "low-frequency background" before "sharpening details." But this seems arbitrary. Have you tried differential→integral ordering? What about applying them in parallel and combining the outputs? The paper would be much stronger with ablations showing this specific sequential ordering is actually important rather than just one choice that happened to work.

3. **Can you provide empirical verification of your theoretical assumptions?** Theorem 1 assumes the Transformer Jacobian has norm ≤1 (assumption A1) and the constrained CNN Jacobian is strongly dissipative with eigenvalues having real part ≤-c (assumption A2). Can you actually measure these quantities for your trained models? Plot the eigenvalue distributions? Without this verification, the theorem just says "if these nice properties hold, then stability follows" but doesn't prove DINO actually achieves them.

4. **How does DINO scale to 3D turbulence problems?** All your experiments are 2D, but real turbulence is fundamentally 3D with different cascade properties. The Transformer's quadratic complexity in the number of spatial points seems like it would make this prohibitively expensive for 3D—a 128³ grid has ~2 million points. Can you show results on full 3D Navier-Stokes, or at least provide a complexity analysis explaining why this would or wouldn't work?

5. **What's the training cost compared to baselines?** You mention DINO starts with a much larger frequency basis during training (1000 modes vs 32 for standard FNO), and Transformers are expensive. Can you report actual training times, memory usage, and wall-clock costs? For practitioners deciding whether to use DINO, knowing that it's 5x or 10x more expensive to train would be really important, even if inference is faster after pruning. Also, what's the inference time comparison after the model is trained is DINO actually faster than FNO for deployment?

---

### Official Review · Reviewer_wQJQ · 2025-10-27

**Soundness:** 3
**Presentation:** 3
**Contribution:** 1
**Rating:** 2
**Confidence:** 4

**Summary:**

This paper proposes the Differential-Integral Neural Operator (DINO) for long-term turbulence forecasting. The authors propose to combine a learnable differential operator layer with a global integral layer to learn a “physics decomposition” of the evolution operator. The authors show that this improves performance on long-term turbulence forecasting for several systems, including the 2D Kolmogorov Flow.

**Strengths:**

The proposed method is well-motivated, and the presentation is clear. I appreciate the detailed experimental comparisons. For example, the authors compare the spectra of the model’s Kolmogorov flow forecasts, which is useful in analyzing the spectral error of each model. I also appreciate the introduction of Geo-DINO at the end. This is an interesting and novel improvement of the architecture, and it adds significantly to demonstrating the architecture’s capabilities.

**Weaknesses:**

The authors may overclaim the novelty of the architecture. In the introduction, the authors write that they “design DINO, a novel neural operator framework that, for the first time, integrates a differential operator with rigorous mathematical convergence guarantees (via constrained CNNs) and a powerful global integral operator (via a Transformer) into a unified parallel model” (page 2). However, [1] below proposes the differential operator via constrained CNNs and combines it in parallel with a global integral operator as well. The primary difference between DINO and LocalNO (from [1]) appears to be that DINO uses a Transformer-based integral kernel, and LocalNO additionally includes a local integral kernel in parallel with the differential and global integral kernels. It is also important to add that adapting self-attention for function spaces has also been mentioned and analyzed in prior works, e.g. [2] and [3].

Furthermore (and please correct me if I am mistaken), the authors do not compare against LocalNO in their experiments. I think this is an important comparison to make, particularly given that the architectural motivation for [1] parallels many of the arguments put forth by the authors in this work.

While the proposed architecture does appear to improve performance on long-range turbulence forecasting, I suggest that the authors consider comparing with or reframing the contributions of the paper in light of some of the prior works discussed here. However, it is certainly possible that I have missed something in my analysis and reading of the paper. If the authors feel this is the case, please feel free to clarify and discuss further.

**Minor notes:**
The theorem in Section 3.5 is worded quite vaguely. I recommend the authors mention that Theorem 1 is meant to be less formal, and I suggest they direct readers to the appendix for a more formal treatment.

**References:**
1. “Neural Operators with Localized Integral and Differential Kernels,” 2024.
2. “Continuum Attention for Neural Operators,” 2025.
3. “Neural Operator: Learning Maps Between Function Spaces,” 2021.

**Questions:**

1. What is the number of parameters for each architecture compared? Were ablations performed for fixed parameter counts?
2. Did the authors perform cross-resolution/super-resolution experiments or experiments at different grids to evaluate this capability of the proposed architecture?
3. As I understand, equation 5 shows the DINO layer. The authors mention that the global integral kernel tends to have an over-smoothing effect. In this case, how does high frequency information propagate to the differential kernel if it must be first passed through the global integral kernel?

---

### Official Review · Reviewer_68dG · 2025-10-28

**Soundness:** 2
**Presentation:** 3
**Contribution:** 2
**Rating:** 2
**Confidence:** 4

**Summary:**

The paper introduces the DINO, a neural operator targeted for long-term forecasting of turbulent physical systems, particularly targeting the major challenge of error accumulation and loss of physical fidelity in existing neural operator models.

The core innovation is a model architecture that mirrors the mathematical structure of the governing partial differential equations (PDEs), specifically the Navier-Stokes equations, by decomposing the network into two synergistic branches: a local differential operator (using a provably convergent convolutional neural network) and a global integral operator (using a Transformer-based self-attention mechanism)

The main concern with this paper is that it presents its results and contributions as a fundamentally new approach, whereas in reality, the work is largely incremental. This in itself would not be an issue, if the conceptual origin of ideas was better communicated

**Strengths:**

- The paper is well written and presents results in a clear manner.
- Code is openly available and the validation seems comprehensive enough in terms of problem settings

**Weaknesses:**

- Despite citing prior work with similar decompositional philosophy, the paper frames its architecture as the first "from-principles" implementation of physics-decomposition in neural operators, whereas prior art laid much of the conceptual and methodological foundation (including moment-constrained kernels and parallel operator branches) [1]. In particular, this approach is very similar to [1], which is cited but not given enough credit, considering that they proposed the idea of splitting the operator and also targeted learning differential and integral parts separately. The main difference here is that a transformer is chosen rather than a spectral (global convolution) to learn the diffusive part. The language could be better nuanced to position this work as a strong empirical validation of the principle, rather than its conceptual origin. I strongly suggest the authors adapt the messaging here.
- Other papers such as [2] are also relevant in this context but not cited.
- Similarly, a number of papers were concerned with stability and have found solutions that remain autoregressively stable. In the weather and climate context, architectures such as SFNO, LUCIE and FCN3 have shown good autoregressive stability [3, 4].
- There is no targeted ablation or head-to-head comparison with [1] specifically. The paper benchmarks against a wide set of alternative operator learning and deep vision models but does not isolate the benefit of their specific transformer-vs-spectral or constrained CNN-vs-standard hybrid relative to. This makes it hard to attribute improvements directly to the differences claimed as novel in this work.
- The approach is moderately novel, given that both transformer neural operators and the differential neural operators already exist and given that this is a slight modification w.r.t. [1]. I wouldn't see this as a problem, if the relation to [1] was more clearly communicated.

[1] Miguel Liu-Schiaffini, Julius Berner, Boris Bonev, Thorsten Kurth, Kamyar Azizzadenesheli, and
Anima Anandkumar. Neural operators with localized integral and differential kernels. arXiv
preprint arXiv:2402.16845, 2024.

[2] Zappala, E., Fonseca, A.H.d.O., Caro, J.O. et al. Learning integral operators via neural integral equations. Nat Mach Intell 6, 1046–1062 (2024). https://doi.org/10.1038/s42256-024-00886-8

[3] Bonev, B., Kurth, T., Mahesh, A., Bisson, M., Kossaifi, J., Kashinath, K., ... & Keller, A. (2025). Fourcastnet 3: A geometric approach to probabilistic machine-learning weather forecasting at scale. arXiv preprint arXiv:2507.12144.

[4] Guan, H., Arcomano, T., Chattopadhyay, A., & Maulik, R. (2024). Lucie: A lightweight uncoupled climate emulator with long-term stability and physical consistency for o (1000)-member ensembles. arXiv preprint arXiv:2405.16297.

**Questions:**

- is the math regarding the stability region of the transformer part also adaptable to convolutional approaches such as [1]? If so, I would put it in the text, as otherwise readers might be mislead that this is only the case for the transformer approach.

---

### Official Review · Reviewer_gQbS · 2025-10-31

**Soundness:** 3
**Presentation:** 3
**Contribution:** 3
**Rating:** 6
**Confidence:** 4

**Summary:**

This paper introduces a new framework for developing neural operators of partial differential equations. The proposed architecture highlights a global integral operator via self-attention, followed by a local differential operator via constrained convolutions. The method is evaluated on long-time forecasting of chaotic systems, including 2-D Kolmogorov flow, 2-D decaying isotropic turbulence, and Prometheus-T dataset. Across these benchmarks, the approach achieves higher accuracy and improved long-term stability than recent baselines such as the Fourier Neural Operator and PDE Refiner.

**Strengths:**

1. Overall, the paper is well written, although some technical details need further clarification.
2. The proposed method achieves much higher accuracy than previous models, as shown in their table 1.
3. The proposed method seems generic and can be applied to different chaotic systems, in addition to the benchmarks in the paper.

**Weaknesses:**

1. There is a conceptual mistake in their theoretical analysis. In Theorem 1, the authors aim to prove that the spectral radius of the Jacobian is bounded by 1, meaning that all of the eigenvalues of the Jacobian are lower than 1. However, this conclusion is inconsistent with the behavior of chaotic systems, which typically have lots of saddle points where the largest eigenvalue of the Jacobian exceed 1. A canonical example is the Lorenz-63 system with parameters sigma = 10, rho = 28, beta = 8/3, where the origin is a saddle point. Any numerical solver that accurately approximates a chaotic system must necessarily have a Jacobian with spectral radius greater than 1 to preserve the system's chaotic behavior.

   Given that the conclusions of Theorem 1 are incompatible with the chaotic systems, which can actually be simulated by their differential-integral neural operator, I suspect that the assumptions underlying Theorem 1 are violated by their approach.

   That being said, the numerical experiments demonstrate that the neural operator achieves stable performance on the systems considered. I suggest the authors either remove Theorem 1, or revise it to prove a result consistent with chaotic systems, e.g. capturing the largest Lyapunov exponent.

2. In Section 4, the authors do not specify architectural details of the baseline models, such as network depth and width. Without this information, it is unclear whether the comparisons are fair or whether performance differences may be attributed to differences in model capacity rather than methodology.

**Questions:**

1. Regarding the Prometheus benchmark, how different are the parameters of testing dataset from the ones used for training? How do they ensure the testing cases are out of distribution of the training datasets?
2. In the sparse ocean forecasting example, how data are used for training / testing?
3. There is a discrepancy between figure 6 and the text: the maximum wind speed marked in figure 6 is 1.47 m/s, while the text reports 1.17 m/s. In addition, according to the colormap of speed errors in figure 6, the maximum error appears to be 1.0~2.0m/s, which is almost 100% relative error with respect to the ground truth. The authors should verify whether there is an error in the colormap labels or provide further explanation for these large errors.

---

### Meta-Review · Area_Chair_qKVW · 2026-01-13

**Summary:**

Reviewers generally indicate that:
1. The theoretical analysis may contain mistakes or make unjustifiable assumptions.
2. The architecture is very similar to prior work but little credit is given and it difficult to gauge the novelty.
3. Important baselines such as LocalNO are not included in the comparisons.
4. Not enough information is given on training details and architectural hyper parameters.

**Reviewer Concerns:**

Authors did not address any reviewer concerns.

**Reviewer Scores:**

Reviewer scores will not change.

---

### Decision · Program_Chairs · 2026-01-26

Reject